# SARS-CoV-2 infection perturbs enhancer mediated transcriptional regulation of key pathways

**Yahel Yedidya, Daniel Davis, Yotam Drier** [ID] *

The Lautenberg Center for Immunology and Cancer Research, IMRIC, Faculty of Medicine, Hebrew University of Jerusalem, Israel

* yotam.drier@mail.huji.ac.il

**Data Availability Statement:** All relevant data are within the manuscript and its Supporting information files.

**Funding:** This research was supported by the Israel Science Foundation (grant 3650/20) and the

## Abstract

Despite extensive studies on the effects of SARS-CoV-2 infection, there is still a lack of understanding of the downstream epigenetic and regulatory alterations in infected cells. In this study, we investigated changes in enhancer acetylation in epithelial lung cells infected with SARS-CoV-2 and their influence on transcriptional regulation and pathway activity. To achieve this, we integrated and reanalyzed data of enhancer acetylation, *ex-vivo* infection and single cell RNA-seq data from human patients. Our findings revealed coordinated changes in enhancers and transcriptional networks. We found that infected cells lose the WT1 transcription factor and demonstrate disruption of WT1-bound enhancers and of their associated target genes. Downstream targets of WT1 are involved in the regulation of the Wnt signaling and the mitogen-activated protein kinase cascade, which indeed exhibit increased activation levels. These findings may provide a potential explanation for the development of pulmonary fibrosis, a lethal complication of COVID-19. Moreover, we revealed over-acetylated enhancers associated with upregulated genes involved in cell adhesion, which could contribute to cell-cell infection of SARS-CoV-2. Furthermore, we demonstrated that enhancers may play a role in the activation of pro-inflammatory cytokines and contribute to excessive inflammation in the lungs, a typical complication of COVID-19. Overall, our analysis provided novel insights into the cell-autonomous dysregulation of enhancer regulation caused by SARS-CoV-2 infection, a step on the path to a deeper molecular understanding of the disease.

## Author summary

The COVID-19 pandemic had a devastating impact, claiming the lives of millions of patients and affecting a large portion of the global population. Despite significant research efforts to investigate the cellular effects of SARS-CoV-2 infection, our understanding of how infection affects cis-regulatory elements remains limited. In this study, we employed computational analysis for integrative analysis of available data of enhancer acetylation and gene expression changes, allowing us to explore the impact of SARS-CoV-2 on cis-regulatory mediated transcriptional regulation. Our findings shed light on the specific

Edmond de Rothschild Foundation (YD). The funders had no role in study design, data collection and analysis, decision to publish, or preparation of the manuscript.

**Competing interests:** The authors have declared that no competing interests exist.

transcription factors and pathways that are affected, providing valuable insights into the relevant to changes associated with the disease.

## Introduction

The COVID-19 pandemic, which emerged in December 2019 has had a profound global impact, resulting in millions of deaths and affecting the lives of countless individuals. This devastating disease is caused by the SARS-CoV-2 virus, and displays a range of clinical manifestations, varying from asymptomatic cases to severe illness. Viruses evolved mechanisms to manipulate their host cells, to ensure their maintenance, replication, and transmission. These manipulations include regulation of the cell cycle, control of apoptosis, and manipulation of the immune system. Since epigenetic modifications play a pivotal role in regulating cellular gene expression, some viruses use epigenetic mechanisms to manipulate the regulation of the host's and their own genes during infection [1–3].

Tremendous recent efforts to unravel the molecular effects of SARS-CoV-2 infection have led us to an improved understanding of the changes in gene expression and biological processes that contribute to disease severity. Investigations into SARS-CoV-2 infection demonstrated large-scale chromatin structural changes which redistribute active and inactive chromatin. Moreover, infection causes a reduction in the overall levels of histone 3 lysine 27 acetylation (H3K27ac), a key marker of active enhancers [4,5]. Other studies have revealed the virus's epigenetic impact on monocytes, a crucial component of the immune response [6]. Previous studies on SARS viruses [7] highlighted the value of epigenetic and regulatory analyses to improve understanding of the disease and devise effective strategies to combat it. However, studies of how SARS-CoV-2 infection affects histone modifications and enhancer regulation in infected lung epithelial cells, which genomic regions are affected, are still lacking, and hold great promise to further uncover mechanisms of pathogenesis and suggest novel therapies.

In this study, we focused on elucidating the cell-autonomous impact of SARS-CoV-2 infection on enhancer regulation in lung epithelial cells. To achieve this, we performed a comprehensive reanalysis and integration of publicly available data on changes in gene expression and H3K27 acetylation following SARS-CoV-2 infection to understand the impact of the infection on enhancer activity and gene regulation. Our findings reveal the upregulation of genes involved in cell adhesion and pro-inflammatory cytokine production, alterations of transcription factor (TF) networks, and differential acetylation of corresponding enhancers. By shedding light on the specific mechanisms through which SARS-CoV-2 infection impacts enhancer regulation in lung epithelial cells, our study contributes to a deeper understanding of the disease pathogenesis and highlights potential targets for therapeutic interventions.

## Results

### Many TFs are significantly enriched at the differential acetylation peaks

We reanalyzed data from an H3K27ac ChIP-seq experiment performed on A549-ACE2 cells 8 hours and 24 hours after SARS-CoV-2 infection (see Methods). We found that 8 hours post-infection there were 744 putative enhancers that gained H3K27 acetylation, and 514 putative enhancers whose acetylation was decreased (FDR<0.05, fold-change > 150%). At 24h post-infection we found 3773 enhancers gaining acetylation and acetylation loss in 8741 enhancers (Fig 1A and S1 Table).

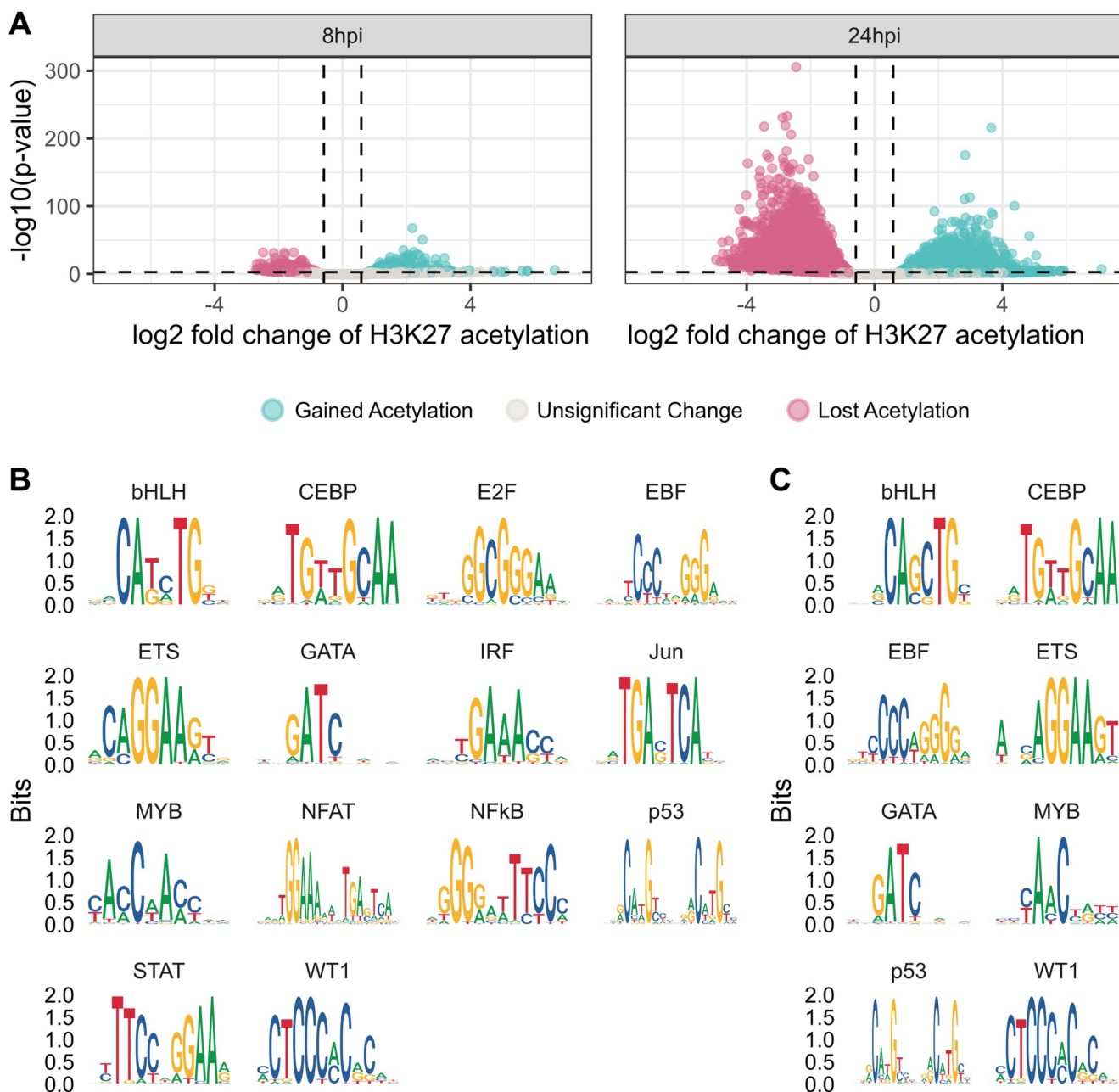

**Fig 1. Changes in enhancer acetylation after SARS-CoV-2 infection.** (A) Ratio of H3K27 acetylation (in log scale) between infected and uninfected A549-ACE2 cells after 8 hours (left) and 24 hours (right). (B) Position weight matrices of transcription factor families enriched in enhancers that significantly gained acetylation. (C) Position weight matrices of transcription factor families enriched in enhancers that significantly lost acetylation.

To understand which transcription factors may be involved in the differentially acetylated enhancers, we performed motif enrichment analysis of regions displaying differential H3K27ac activity for each time point (FDR < 0.05). In enhancers that gained H3K27ac, we found an enrichment of motifs recognized by nuclear factor κB (NF-κB), GATA family, STAT family, IRF family, and EBF family, which are related to immune response. In addition, we found enrichment of motifs recognized by AP-1, p53, E2F, ETS, bHLH, and MYB (Fig 1B and

**Table 1. Transcription factors whose binding is enriched (FDR<0.05) in enhancers that lose acetylation or gain acetylation (FDR<0.05, fold-change > 150%) 8h and 24h after infection.**

| | 8h post-infection | 24h post-infection |
|---|---|---|
| Lower acetylation levels | EP300, HES2, CEBPB | MAFK, EP300, SMC3, HES2, JUN, TP63, EHMT2, GATA3, NR3C1, CHD4, KDM1A, MAX, RAD21, TEAD4, CEBPB, JUNB, FOSL2 |
| Higher acetylation levels | EP300, GATA3, FOSL2, JUNB | EHMT2 |

S2 Table). In enhancers that lost their acetylation, we found enrichment of motifs from the bHLH family, ETS family, and MYB family (Fig 1C and S2 Table).

To identify the actual TFs that may bind the differentially acetylated enhancers, we obtained the binding sites of 27 DNA binding factors experimentally measured in A549 cells from ENCODE. Binding sites of 17 out of these factors were found to be significantly enriched in enhancers that lost acetylation after 24h, and a few of them were also found to be enriched in enhancers that lost or gained acetylation after 8h (Fisher exact test, FDR < 0.05, Table 1). The 17 enriched factors include factors related to transcriptional machinery and epigenetic regulation (EP300, EHMT2, SMC3, KDM1A, RAD21, TEAD4, CHD4, FOSL2, JUNB, JUN) and immune-related processes (CEBPB, GATA3, MAFK, MAX, NR3C1). Notably, for MAFK (acetylation loss after 24h), JUNB (acetylation gain after 8h), GATA3 (acetylation gain after 8h), and MAX (acetylation loss after 24h), the binding sites enrichment is also supported by a matching motif enrichment.

## Differentially acetylated enhancers are adjacent to differentially expressed genes

To test the correlation of enhancer changes with gene expression and focus on clinically relevant changes, we obtained gene expression data of *ex-vivo* infected human lung tissues and human COVID-19 patients' lungs (see Methods). We looked for significantly differentially expressed genes near enhancers with significant acetylation change. For lungs infected *ex-vivo*, we found that 87 genes that were differentially expressed 24h after SARS-CoV-2 infection compared to mock control (FDR < 0.1, fold-change > 150%, Fig 2A) and that were located near enhancers with significant acetylation change (FDR < 0.05, fold-change > 150%). 80 were more highly expressed, and 7 genes had lower expression. We found that enhancers that lost acetylation tend to be near genes with significantly lower expression (Fisher exact test, p < 0.04, Fig 2B) and enhancers that gained acetylation tend to be near (< 100kb) genes that had a significantly higher expression (Fisher exact test, p < 0.004, Fig 2B).

When we looked at the distribution of acetylation levels around the transcription start site (TSS) of genes with significant expression change, we found that acetylation was indeed observed mostly at the promoter region upstream of the TSS, and importantly gain of acetylation at promoters was significantly higher in genes that gained expression (t-test, p < 0.03, Fig 2C and 2D).

For the COVID-19 patient lung data, we found that 930 genes had a significant expression change post-infection (FDR < 0.05, fold-change > 150%, Fig 2E) and were located near enhancers with significant acetylation change (FDR < 0.05, fold-change > 150%). 560 genes had a significantly higher expression, and 370 genes had a significantly lower expression. Enhancers that lost acetylation tend to be found near (< 100 kb) genes with significantly lower expression (Fisher exact test, $p < 10^{-6}$, Fig 2F). In this case, no significant enrichment was found between enhancers that gained acetylation and highly expressed genes. Again,

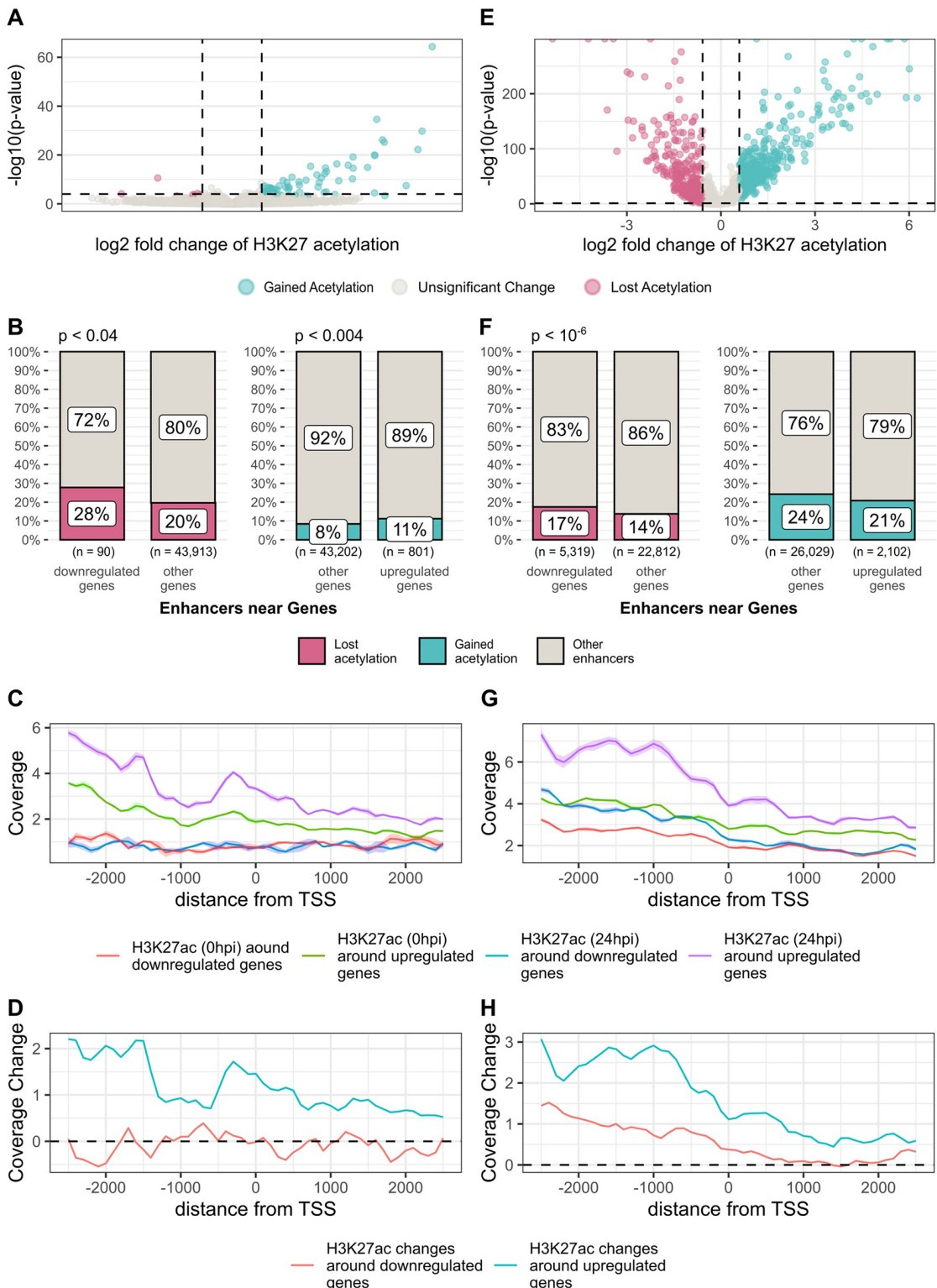

**Fig 2. Correspondence between enhancer acetylation changes and gene expression changes.** (A) Expression fold changes (in log scale) of genes near (<100 kb) differentially acetylated enhancers 24h after *ex-vivo* infection of lung tissues. (B) Association between differentially acetylated enhancers and differentially expressed genes 24h after *ex-vivo* infection of lung tissues. Out of 90 enhancers near (<100 kb) downregulated genes, 25 lost acetylation 24h post infection, a statistically significant overlap (p < 0.04). Out of 801 enhancers near (<100 kb) upregulated genes, 90 gained acetylation 24h post infection, a statistically significant overlap (p < 0.004).

(C) H3K27 acetylation distribution around the transcription start site of differentially expressed genes after *ex-vivo* infection of lung tissues. (D) Fold change of H3K27 acetylation 24h after SARS-CoV-2 infection around differentially expressed genes after *ex-vivo* infection of lung tissues. (E) Expression fold changes of genes near (<100 kb) differentially acetylated enhancers from COVID-19 patients' lungs. (F) Association between differentially acetylated enhancers and differentially expressed genes 24h from COVID-19 patients' lungs. Out of 5319 enhancers near (<100 kb) downregulated genes, 927 lost acetylation 24 h post infection, a statistically significant overlap ($p < 10^{-6}$). Out of 2102 enhancers near (<100 kb) upregulated genes, 438 gained acetylation 24h post infection, no significant enrichment. (G) H3K27 acetylation distribution around the transcription start site of differentially expressed genes from COVID-19 patients' lungs. (H) Fold change of H3K27 acetylation 24 h after SARS-CoV-2 infection around differentially expressed genes from COVID-19 patients' lungs.

promoters gained acetylation mostly at genes that gained expression after SARS-CoV-2 infection (t-test, p < 0.04, Fig 2G and 2H).

## Acetylation changes linked to cell adhesion and inflammatory response pathways

To understand which biological processes are affected by changes in enhancer activity, we performed gene set enrichment analysis for differentially expressed genes near differentially acetylated enhancers. We analyzed differentially expressed genes separately for *ex-vivo* infected lung tissues and COVID-19 patients' lung data. Several processes were consistently enriched between the COVID-19 patient data and the lung tissues data, including genes involved in the regulation of cell adhesion, positive regulation of ERK1 and ERK2 cascade, positive regulation of NF-kappaB signaling, and many pathways related to immune processes, including interferon signaling (Fig 3 and S3 and S4 Tables).

## Association with Wnt signaling, MAPK signaling, NF-kappaB and adhesion

Since the gene set enrichment analysis indicated enrichment in Wnt signaling, MAPK signaling, NF-kappaB signaling, and cell adhesion, we further examined the differential genes (FDR < 0.1, fold-change > 150%) located near (< 100kb) differentially acetylated enhancers and involved in these pathways (S5 Table, Fig 4A). Due to a lower statistical power to detect differential expression in the *ex-vivo* data, we relied on the COVID-19 patient data for this analysis. Overall, we saw a loss of negative regulators and gained expression of positive regulators of adhesion, NF-kappaB, and Wnt signaling, while for MAPK signaling changes in expression were more complex and not in a clear direction (Table 2).

For the Wnt signaling pathway we found upregulation of positive regulators, such as BCL9L, CDH3, CSNK1E, and SDC1, as well as downregulation of negative regulators, such as CAV1, PLPP3, and SFRP4 (S5 Table). For NF-kappaB signaling, we found upregulation of many positive regulators and of the TF subunits: APP, CASP10, EEF1D, EIF2AK2, IRAK3, MAVS, NPM1, RPS3, and TRIM22 as well as downregulation of negative regulators, such as CTNNB1, PELI1, and RORA (S5 Table). Again, we also found downregulation of positive regulators and subunits: AKAP13, CAV1, CD36, MAP3K14, PELI2, and REL and upregulation of negative regulators: TRIM38, TNIP1, and HSPB1. For MAPK signaling pathway we found upregulation of positive regulators and genes that are involved in the MAPK cascade: ADAM9, APP, EIF2AK2, GAS6, KITLG, GDF15, NOTCH1, and TRIM5, and downregulation of negative regulators: CAV1, CNKSR3, DUSP1, DUSP6, PDE8A, SPRED2, and PELI2. In a few cases changes in expression did not support MAPK activation when positive activators, such as AKAP13, PLCE1, RIPK1, and AREG, were downregulated and negative regulators like STK38 and TNIP1 were upregulated (S5 Table). Together this suggested that SARS-CoV-2

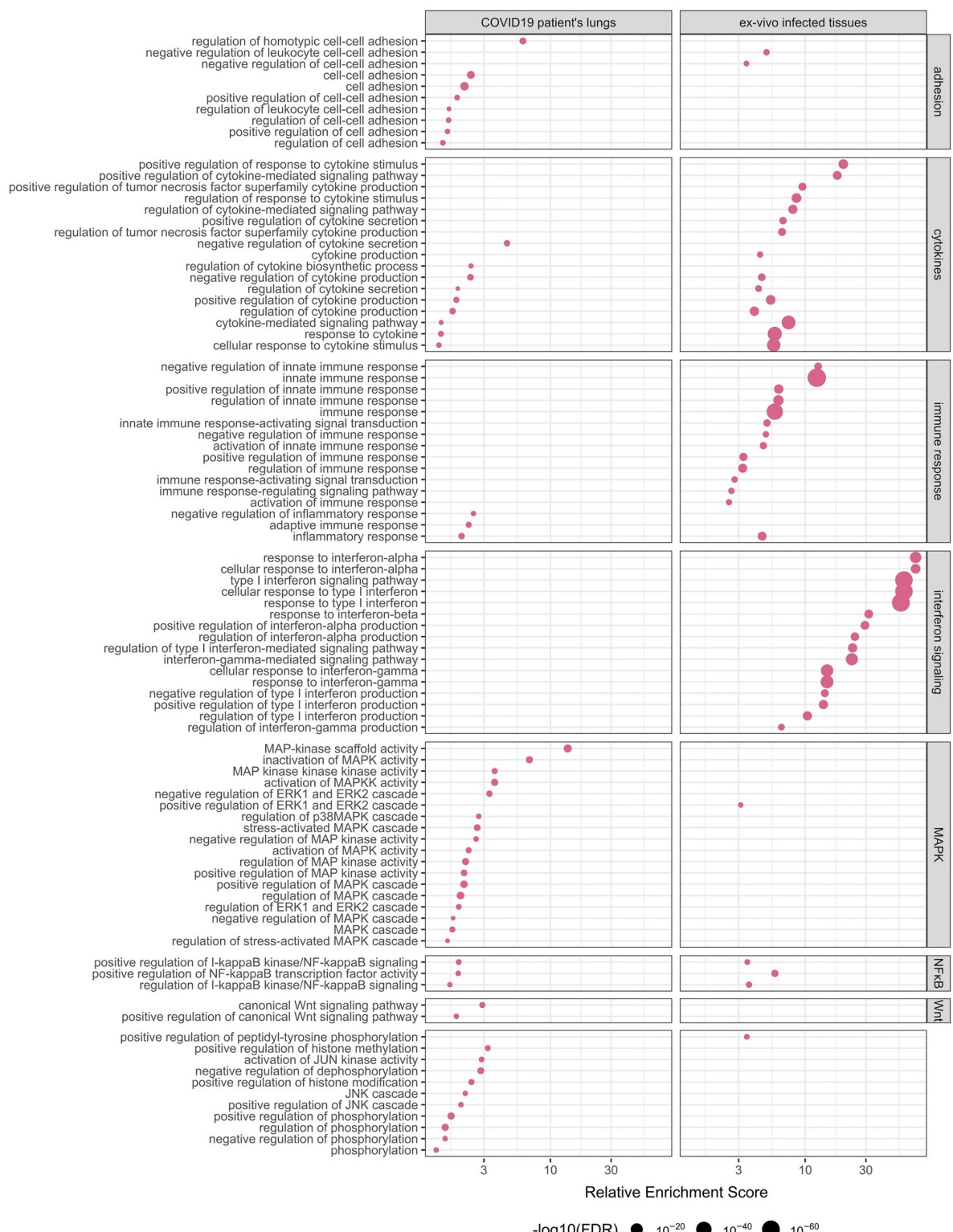

**Fig 3. Gene set enrichment analysis of genes that are significantly differentially expressed (fold-change > 150%, FDR < 0.1) and are near differently acetylated enhancers (fold-change > 150%, FDR < 0.05), only significant pathways (FDR < 0.1, target gene in term > 2) are shown.** Size of the dot represent false discovery rate (FDR).

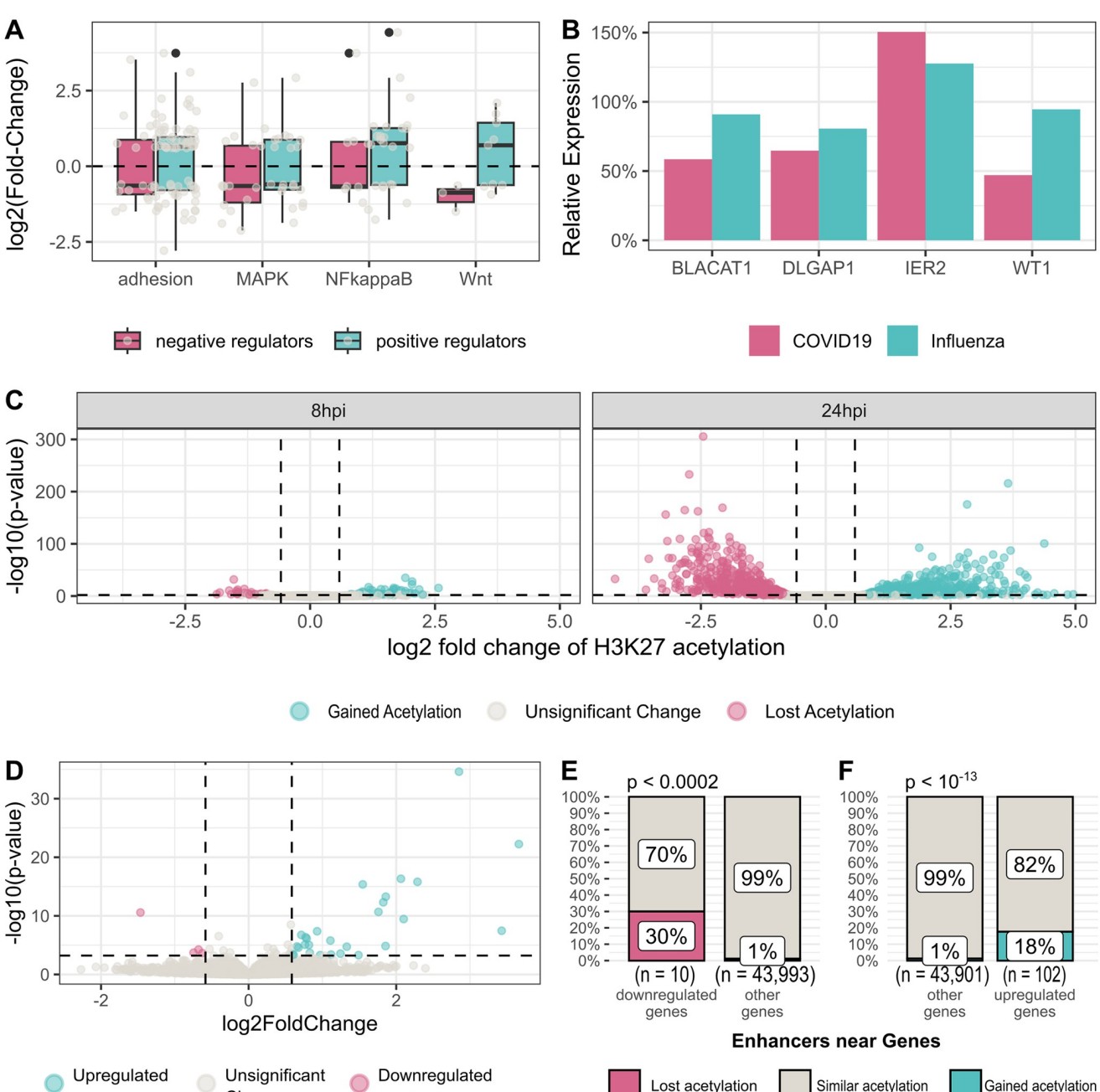

**Fig 4. Enhancer mediated regulation of Wnt signaling, MAPK signaling, NF-kappaB signaling and cell adhesion is perturbed by SARS-CoV-2 infection and WT1 loss.** (A) Distribution of the fold change in the expression of significant genes in the mentioned pathways. (B) Gene expression changes of the four genes that are differentially expressed after SARS-CoV-2 infection but not after influenza infection. Bar plot show fold-change of expression before and 24h after infection in log scale. (C) Change of H3K27 acetylation at WT1 enhancers in A549-ACE2 cells 8 and 24h after SARS-CoV-2 infection compared to uninfected cells. (D) Expression changes of genes near (< 100kb) differentially acetylated WT1 enhancers 24h after *ex-vivo* infection of lung tissues. (E) Out of 10 downregulated genes after *ex-vivo* infection of lung tissues, 3 are near (< 100kb) WT1 enhancers that lose acetylation 24h post infection, a statistically significant overlap (p < 0.0002) (F) Out of 102 upregulated genes after *ex-vivo* infection of lung tissues, 18 are near (< 100kb) WT1 enhancers that gain acetylation, a statistically significant overlap (p < $10^{-13}$).

**Table 2. Differentially expressed genes (fold-change > 150%, FDR < 0.1, 24h post-infection, COVID-19 patient lung data) that act as positive/negative (+/-) regulators involved in Wnt signaling, NF$\kappa$B signaling, MAPK signaling, or cell adhesion pathways.**

| | Regulator | log2(fold-change) | Wnt signaling | NF-$\kappa$B signaling | MAPK signaling | Cell adhesion |
|---|---|---|---|---|---|---|
| upregulated genes (fold-change > 150%, FDR < 0.1) | BCL9L | 1.437 | + | | | |
| | CDH3 | 1.733 | + | | | |
| | CSNK1E | 0.880 | + | | | |
| | SDC1 | 2.091 | + | | | |
| | APP | 0.957 | | + | + | + |
| | CASP10 | 1.000 | | + | | |
| | EEF1D | 2.267 | | + | | |
| | EIF2AK2 | 0.879 | | + | + | |
| | IRAK3 | 0.649 | | + | | |
| | MAVS | 0.612 | | + | | |
| | NPM1 | 1.350 | | + | | |
| | RPS3 | 4.423 | | + | | |
| | TRIM5 | 1.244 | | | + | |
| | TRIM22 | 1.300 | | + | | |
| | TRIM38 | 0.785 | | - | | |
| | TNIP1 | 0.827 | | - | - | |
| | HSPB1 | 3.737 | | - | | + |
| | ADAM9 | 0.867 | | | + | |
| | KITLG | 0.639 | | | + | |
| | GAS6 | 0.907 | | | + | + |
| | STK38 | 0.987 | | | - | |
| | GDF15 | 2.764 | | | + | |
| | NOTCH1 | 0.662 | | | + | |
| | ACTG1 | 2.233 | | | | + |
| | BCAM | 1.613 | | | | + |
| | CD63 | 1.970 | | | | + |
| | CYTH3 | 0.635 | | | | + |
| | EPCAM | 0.886 | | | | + |
| | FLOT1 | 0.762 | | | | + |
| | JUP | 1.223 | | | | + |
| | PML | 1.235 | | | | + |
| | FRMD5 | 1.823 | | | | + |
| | PPP1CB | 0.606 | | | | + |
| | PTK2 | 0.686 | | | | + |
| | PTK7 | 1.113 | | | | + |

(*Continued*)

**Table 2.** (Continued)

| | Regulator | log2(fold-change) | Wnt signaling | NF-κB signaling | MAPK signaling | Cell adhesion |
|---|---|---|---|---|---|---|
| downregulated genes (fold-change > 150%, FDR < 0.1) | CAV1 | -0.642 | - | + | - | - |
| | PLPP3 | -1.496 | - | | | |
| | SFRP4 | -0.875 | - | | | |
| | CTNNB1 | -0.666 | | - | | |
| | RORA | -0.748 | | - | | |
| | PELI1 | -1.204 | | - | | |
| | PELI2 | -0.646 | | + | - | |
| | AKAP13 | -0.597 | | + | + | |
| | CD36 | -1.434 | | + | | |
| | MAP3K14 | -1.218 | | + | | |
| | REL | -0.693 | | + | | |
| | CNKSR3 | -1.286 | | | - | |
| | DUSP1 | -1.113 | | | - | - |
| | DUSP6 | -2.123 | | | - | |
| | PDE8A | -0.652 | | | - | |
| | SPRED2 | -0.703 | | | - | |
| | PLCE1 | -1.396 | | | + | |
| | RIPK1 | -0.617 | | | + | |
| | AREG | -1.869 | | | + | |
| | DLG2 | -1.475 | | | | - |
| | KNAK1 | -0.598 | | | | - |
| | NINJ2 | -0.794 | | | | - |
| | PARVA | -0.823 | | | | - |
| | PTPRS | -0.686 | | | | - |
| | IGFBP7 | -0.700 | | | | - |

infection can trigger Wnt and NF-kappaB signaling, promoting both the expression of activators and the silencing of repressors of these pathways.

For cell adhesion, we found upregulation of positive regulators and genes that are involved in adhesion processes, such as ACTG1, APP, BCAM, CD63, CDH3, CYTH3, EPCAM, FLOT1, FRMD5, GAS6, HSPB1, JUP, PML, PPP1CB, PTK2, and PTK7 and downregulation of adhesion negative regulators, such as CAV1, DLG2, DUSP1, KANK1, NINJ2, PARVA, PTPRS, and IGFBP7 (S5 Table). This suggested that SARS-CoV-2 infection can promote cell adhesion by inducing positive regulators and repressing negative regulators. Adhesion may play an important role in cell-to-cell infection, a major infection route for these SARS-CoV-2 strains, and may be an important mechanism by which SARS-CoV-2 manipulates the cells to increase its ability to infect neighboring cells. Several of these genes (PELI1, MAP3K8, LAMA3, EMP2, CTNNAL1, CAV1, LRRK2, SFRP4) may also be involved in lung fibrosis, a severe complication of COVID-19.

## WT1 is enriched in H3K27ac peaks that lose acetylation after covid infection

To focus on SARS-CoV-2-specific effects, we repeated the analysis but removed genes that are also modulated by influenza infection (fold-change > 150%, FDR < 0.1), relying on the *ex-vivo* data, which provided matched SARS-CoV-2 and influenza control at 24 h. Only four

genes passed this criterion: IER2, DLGAP1, WT1, and BLACAT1. All these genes were located near (< 100kb) enhancers that lose acetylation. WT1, DLGAP1, and BLACAT1 were also significantly downregulated at this time point (Fig 4B).

Since WT1 (Wilms Tumor Suppressor) is a transcription factor that is downregulated during SARS-CoV-2 infection but not during influenza infection, we focused on the transcriptional effects of its downregulation. The WT1 gene encodes a zinc finger transcription factor and RNA-binding protein that directs the development of several organs and tissues. WT1 regulates numerous target genes that are involved in growth, differentiation, and apoptosis. It can serve both as a transcriptional activator and a suppressor. As mentioned above (Fig 1B and 1C), we found that the motif recognized by WT1 is enriched both at enhancers that lose acetylation and at enhancers that gain acetylation, suggesting WT1 indeed serves both as an activator and a repressor in this context as well.

We identified which of the enhancers contains a WT1 motif ("WT1 enhancers"). The WT1 motif was found at 85 of the differentially acetylated enhancers 8h after infection (FDR<0.05, fold-change>150%). 56 of those significantly gained acetylation, and 29 lost acetylation. At 24h post-infection, the WT1 motif was found at 1118 differentially acetylated enhancers (FDR<0.05, fold-change>150%)—595 of those gained acetylation and 523 lost acetylation (Fig 4C). We found 45 differentially expressed genes (*ex-vivo* infected data, 24h post-infection, FDR < 0.1, fold-change > 150%) located within 100 kb of differential WT1 enhancers—37 upregulated genes and 8 downregulated genes (Fig 4D).

We found 5 downregulated genes near WT1 enhancers that lost acetylation: DLGAP1, MAVS, EPHX1, PXMP4, and SPN. Finally, we found that WT1 enhancers that lost their acetylation tend to be found next to significantly downregulated genes (Fisher exact test, $p < 0.0002$, Fig 4E) and that WT1 enhancers that gained acetylation tend to be found next to upregulated genes (Fisher exact test, $p < 10^{-13}$, Fig 4F).

To better understand the function of WT1 targets in this context, we performed gene set enrichment analysis for differentially expressed genes (SARS-CoV-2 *vs.* mock, fold-change 150%, FDR<0.1) near (< 100kb) differentially acetylated WT1 enhancers. We found enrichment of genes related to regulation of MAPK cascade, phosphorylation, cell adhesion, cytokine production, and other immune processes (Fig 5, S6 Table).

Together, this data suggested that WT1 was down-regulated after SARS-CoV-2 infections, which leads to major changes in the acetylation of WT1 bound enhancers, and the transcriptional regulation of numerous targets. This process was specific for SARS-CoV-2 and is not shared with other respiratory infections such as influenza, may be responsible for some of the unique features of COVID-19 such as excessive acute inflammation, and may support cell-to-cell infection.

## Discussion and conclusion

The COVID-19 pandemic affected the lives of most of humanity in recent years. Despite record breaking efforts to reveal the molecular impact of SARS-CoV-2 infection, our understanding of the impact on enhancer mediated transcriptional regulation remains poor. Given the crucial role of enhancer regulation and chromatin modifications in the regulation of transcriptional networks and cellular gene expression, our study aimed to elucidate the impact of SARS-CoV-2 infection on H3K27ac, a histone modification associated with active enhancers, and downstream gene expression changes.

Our findings revealed substantial changes in enhancer acetylation at 8 and 24 hours postinfection. Both gain and loss of acetylation were common and associated with differential expression of nearby genes. While H3K27ac data is currently limited to *in vitro* data due to

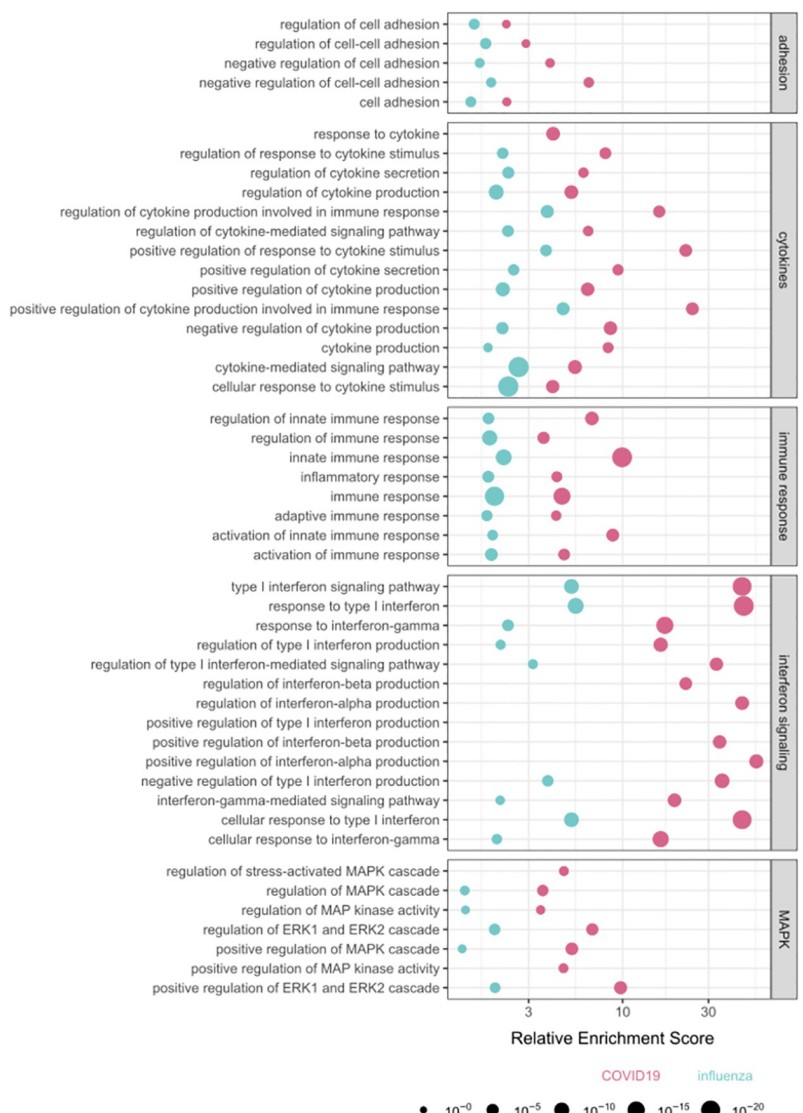

**Fig 5. Gene set enrichment analysis of genes that are significantly differentially expressed (fold-change > 125%, FDR < 0.1, target genes in term > 2) and are near differently acetylated WT1 enhancers (fold-change > 150%, FDR < 0.05), only significant pathways (FDR < 0.1, target gene in term > 2) are shown.** Size of the dot represent false discovery rate (FDR).

technical difficulties, we demonstrated here its relevance to human biology by establishing a significant agreement with gene expression data in COVID-19 patients and *ex-vivo* infected lung samples. Furthermore, our analysis of motif and TF binding data at differential enhancers identified several transcription factors whose activity underwent significant alterations. While some of these effects are related to the general immune response of the infected cells, we identified several changes that are unique to SARS-CoV-2 and were not observed after influenza infection, most prominently downregulation of WT1 and its effect on WT1 bound enhancers and their target genes.

We discovered that differentially acetylated enhancers are adjacent to differentially expressed genes. Gene set enrichment analysis revealed changes in the inflammatory response,

cell adhesion, Wnt signaling, MAPK cascade, and NF-kappaB signaling. In particular, careful examination revealed the upregulation of positive regulators and the downregulation of negative regulators of Wnt signaling, NF-kappaB signaling, and cell adhesion, suggesting coordinated regulation to activate these pathways. While previous studies have already uncovered SARS-CoV-2 infection activates Wnt [8–10], MAPK [11–13], and NF-kappaB [11,14,15], our research unveils the involvement of enhancer mediated transcriptional networks in these processes for the first time. Furthermore, the activation of cell adhesion suggests that SARS-CoV-2 infection orchestrates transcriptional networks to support cell-to-cell transmission, a critical infection route for some SARS-CoV-2 strains, including those investigated in our study [16].

Importantly, our investigation demonstrated significant downregulation of WT1 during SARS-CoV-2 infection, and that this is not part of the general response observed in respiratory infections as it does not occur in influenza infections. WT1 is known as a negative regulator of MAPK signaling pathway [17], acting through the activation of SPRY1 [18] and DUSP6 (MKP3) [19]. Indeed, differentially expressed genes near WT1 enhancers were enriched in these processes, implying that the loss of WT1 contributes, at least in part, to the activation of the MAPK pathway. The novel finding of WT1's impact on increased cell adhesion warrants further investigation and may be a significant factor in facilitating cell-to-cell infection of SARS-CoV-2.

Our analysis provides valuable novel insights into the cell-autonomous transcriptional effects of SARS-CoV-2 infection, an important step toward a more comprehensive molecular understanding of the disease. Furthermore, our findings generate several hypotheses that merit further exploration and validation. This study showcases the potential of enhancer analysis in uncovering the regulatory effects of SARS-CoV-2 infection, as well as its applicability to other viral infections. Specifically, our analysis revealed the role of enhancer mediated transcriptional regulation after SARS-CoV-2 infection in manipulating WT1 activity, cell-to-cell infection, MAPK signaling, Wnt signaling and pro-inflammatory cytokines. Overall, our work demonstrated how integrative analysis of enhancer acetylation and gene expression after infection can reveal new pathways and transcriptional cascades that are perturbed by SARS-CoV-2 infection, moving us forward toward a deeper understanding of the molecular underpinning of COVID-19.

## Methods

### Analyzed acetylation peaks

Fastq files of H3K27ac ChIP-seq in ACE2-A549 cells, before and after infection with SARS-CoV-2 (strain USA-WA1/2020) reported in [4] were obtained from the GEO database, accession GSE167528. The fastq files were aligned to the human reference genome GRCh38 using bowtie2 version 2.3.4.3 [20] with default parameters. The resulting BAM files were sorted with samtools (version 1.9, [21]). Picard (version 2.26.2, [22]) was employed to eliminate PCR duplicates. Peaks, representing regions of enriched H3K27ac signal, were called using MACS2 [23] with the following parameters: callpeaks -g hs -q 0.01 --broad -c input, to allow identification of broad enhancers. Peaks overlapping ENCODE blacklisted regions [24] were excluded to remove potential artifacts.

A union set of peaks for each comparison (ctrl vs. 8h and ctrl vs. 24h), was generated using bedtools merge [25]. Reads in each peak for each experimental condition were counted using bedtools multicov function. The obtained read counts were utilized as input for DESeq2 (default parameters) [26] to identify differentially acetylated peaks. A cutoff of FDR adjusted p-value < 0.05 was applied to identify significantly different H3K27ac peaks for further analysis. HOMER (version 4.11) was used to identify enriched motifs in the sequences of the

differentially acetylated enhancers. The merged set of enhancer peaks was used as background to control for common motifs in A549 cells and focus on differential motifs.

### TFs enrichment in H3K27ac peaks

ChIP-Seq peaks of 27 transcription factors (TFs) in A549 cells were obtained from ENCODE [27,28]: ENCFF008SZM, ENCFF169BSN, ENCFF307JCM, ENCFF469DNJ, ENCFF652DIB, ENCFF695MMQ, ENCFF786NAO, ENCFF976HNC, ENCFF129FEV, ENCFF209ZVL, ENCFF341HVV, ENCFF495URV, ENCFF598BZD, ENCFF656HFI, ENCFF702XIF, ENCFF808XJN, ENCFF988QUL, ENCFF143OQP, ENCFF280RQS, ENCFF399ZUJ, ENCFF517TKD, ENCFF624ZSR, ENCFF669UYG, ENCFF713RHL, ENCFF814JWH, ENCFF146MNV, ENCFF294QSH, ENCFF455TWM, ENCFF562UOF, ENCFF639RSC, ENCFF693LFI, ENCFF766VUQ, ENCFF894YJQ. To identify TFs associated with significantly differential H3K27ac peaks at specific time point (8h,24h) and in specific direction (gain/loss), we utilized bedtools to intersect the peaks of each TF with the differentially acetylated H3K27ac peaks. Fisher exact test (one-tail) was employed to determine if the transcription factors were significantly enriched in the intersect peaks, using the set of all enhancers as background.

### Gene expression reanalysis

We downloaded gene expression counts of single-cell RNA-seq of COVID-19 patients' lungs [29]. We focused on lung epithelial cells, as specified in the original publication. Cells that expressed less than 1000 transcripts and cells in which more than 20% of the reads map to mitochondrial genes were excluded. Then, genes that were expressed in less than 10% of the remaining cells were excluded. Differentially expressed genes between infected and naïve cells were identified by the Wilcoxon test, and the Benjamini-Hochberg procedure controlled the false discovery rate (FDR).

DESeq2 results of gene expression data of lung tissue *ex-vivo* infected with SARS-CoV-2 or influenza [30] were downloaded from GEO, accession GSE163959. Genes with a fold-change greater than 150% and an FDR below 0.1 were defined as differentially expressed.

### Association between change in enhancer acetylation and gene expression

Fisher exact test (one-tail) was used to determine if enhancers exhibiting higher/lower acetylation were enriched in the set of enhancers near ($< 100$kb) upregulated/downregulated genes. Similarly, Fisher exact test was used to evaluate whether upregulated/downregulated genes were enriched in the set of genes near ($<100$kb) over/under acetylated enhancers. The background for these analyses consisted of all enhancers or genes found within a 100kb distance from other genes or enhancers, respectively, to ensure genes without adjacent enhancers do not bias the analysis.

### Acetylation levels around TSS

To generate the H3K27 acetylation profiles around transcription start sites of differentially expressed genes (Fig 2), we counted H3K27ac ChIP-seq reads at bins of size 100bp using HOMER annotatePeaks. We normalized these results by the size factors computed by DESeq2.

### Gene set enrichment analysis

Gene set enrichment analysis was done with the Homer FindGO tool. We used all genes located near ($< 100$kb) all enhancers served as background for all of the gene set enrichment

analyses. In order to compare the enrichment level of the pathways we calculated the relative enrichment score by: (number of target genes in term * total number of genes)/(total number of target genes * number of genes in term), as suggested in [31].

## WT1 enhancers identification

FIMO tool (version 5.5.0) [32] of MEME suite [33] was used to find enhancers that contain WT1 known motif (MCTCCCMCRCAB). Enhancers that exhibited the presence of the WT1 known motif with false discovery rate (FDR) < 0.1 were classified as WT1 enhancers.

## Supporting information

**S1 Table. Differentially acetylated enhancers 8 h and 24 h after SARS-CoV-2 infection.**
(XLSX)

**S2 Table. HOMER motif analysis results for differentially acetylated enhancers 8 h and 24 h after SARS-CoV-2 infection.**
(XLSX)

**S3 Table. Gene set enrichment analysis results for genes with significant expression change located near differentially acetylated enhancers (COVID-19 patient data).**
(XLSX)

**S4 Table. Gene set enrichment analysis results for genes with significant expression change located near differentially acetylated enhancers (*ex-vivo* infected lung tissue data).**
(XLSX)

**S5 Table. Expression change of genes from cell adhesion, Wnt signaling and MAPK signaling.**
(XLSX)

**S6 Table. Genes set enrichment analysis results for genes with significant expression change near WT1 enhancers.**
(XLSX)

**S7 Table. Expression change of the enriched DNA binding factors 24h after SARS-CoV-2 infection.**
(XLSX)

## Author Contributions

**Conceptualization:** Yotam Drier.

**Formal analysis:** Yahel Yedidya, Daniel Davis.

**Funding acquisition:** Yotam Drier.

**Investigation:** Yahel Yedidya.

**Methodology:** Yahel Yedidya, Yotam Drier.

**Project administration:** Yotam Drier.

**Resources:** Yotam Drier.

**Supervision:** Yotam Drier.

**Visualization:** Yahel Yedidya, Yotam Drier.

**Writing – original draft:** Yahel Yedidya, Daniel Davis, Yotam Drier.

**Writing – review & editing:** Yahel Yedidya, Yotam Drier.

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
