## [Decision Letter · Decision Letter 0]

8 May 2023

Dear Dr. Drier,

Thank you very much for submitting your manuscript "SARS-CoV-2 infection perturbs transcriptional regulation of lung cells, supporting cell-cell infection, and inducing Wnt signaling, MAPK signaling and pro-inflammatory cytokines" for consideration at PLOS Computational Biology.

As with all papers reviewed by the journal, your manuscript was reviewed by members of the editorial board and by several independent reviewers. In light of the reviews (below this email), we would like to invite the resubmission of a significantly-revised version that takes into account the reviewers' comments.

We cannot make any decision about publication until we have seen the revised manuscript and your response to the reviewers' comments. Your revised manuscript is also likely to be sent to reviewers for further evaluation.

Sincerely,

Shailendra Shivaji Gurav, PhD

Guest Editor

PLOS Computational Biology

Rob De Boer

Section Editor

PLOS Computational Biology

Dear Authors,

I have completed my evaluation of your manuscript. The reviewers recommend reconsideration of your manuscript following major revision. I invite you to resubmit your manuscript after addressing the all reviewers comments.

Reviewer's Responses to Questions

**Comments to the Authors:**

Reviewer #1: 1. Title- It should be simple and clear. Consider modifying it.

2. Short Title- The present short title does not seem to convey the message of this study.

3. Keywords- Keep them specific. Avoid general words like ‘epigenetics’. Words from present title can be projected as keywords provided the title is considered to be modified. It would help to improve the citations of this study.

4. Abstract- It should reflect the summary of the study with 1-2 introductory lines, objective statement, brief methodology, key results, and concluding remarks. This draft seems ambiguous that lacks clarity.

5. Abstract- Avoid abbreviations right away (E.g., WT1). Mention abbreviation in a full form at the first place it appears.

6. Introduction- This section reads incompetent to establish the study. It should contain disease burden, problem statement, and present approaches to address it. It also should introduce the biochemical factors and their significance in disease progression. For example- What is the rational of acetylation in COVID-19 pathogenesis? This section can be concluded with a paragraph emphasizing approach or plan of present study.

7. Results- Figure 1(A) claims to depict the change in H3K27 acetylation in infected cells as compared to uninfected cells. In reality, the figure does not show the data points of acetylation in uninfected cells.

8. Results- Figure 2 conveys the association between acetylation gain and gene upregulation. However, sub-figures (B, C, G, and H) can be represented with bar graph after percent normalization for ease of understanding.

9. Results- A table of pathway regulators is recommended instead of mentioning them all in the text.

10. Results- For figure 4 (C and F), refer comment No. 8.

11. Discussion- This section reads conclusion instead of discussion. This section should discuss the obtained results at the background of available literature. It should identify present research gaps and comment how your observations helped to fill those gaps.

12. Conclusion- This section is major missing.

13. Methodology- Explain the rationale behind every method that was followed.

14. Methodology- It contains the word ‘we’ consistently. Rephrase the sentences and avoid first-person narrative.

15. General- Authors are supposed to be well-versed with the research in this area. The study should establish a rationale or hypothesis properly. There is a great scope of improvement in scientific writing, development of logical methodology, data representation and interpretation.

Reviewer #2: Dear Authors,

The research article under consideration is a nice research work which would prove very useful with challenges we are facing with the pandemic and post pandemic issues. However, this manuscript needs some corrections and revisions as listed below.

The title could be simplified for better understanding, may be avoid more technical terms.

The abstract need revision, it need to reflect the contents and the findings.

Introduction and methodology needs to be more comprehensive.

Please correct grammatical and topological errors.

A takeaway message in form of conclusion would be very helpful.

Thank you

Reviewer #3: Dear authors, I found your draft quite interesting to read and I really loved doing this. However, there are some concerns I couldn’t accept in the present form. I am summarizing few of them below; I hope this would help you for further refinement of your draft.

• Statement “Despite extensive studies of the effects of SARS-CoV-2 infection, we still lack an understanding of the downstream epigenetic and transcriptional alterations” ; however, few information is available- (doi: 10.1016/j.virusres.2022.198853; https://doi.org/10.1186/s13073-022-01137-4;
https://doi.org/10.1186/s13148-020-00946-x) I think you need to compare your findings with these evidences

• Figure 1 (b and c) can be revised. I feel you have provided the default images; however, it looks inadequate and can be presented in simpler way

• In table you mention “lower and higher acetylation levels”; what defines this? Is there any threshold to mention whether the acetylation is higher or lower?

• Please avoid providing any references in the results section

• All the Venn diagrams should be revised; the area containing higher items should be larger than another; please revise

• I see your study focuses towards Wnt signaling, MAPK signaling and pro-inflammatory which are also regulated by IFN1. Just out of curiosity, did you found any epigenetic and transcriptional alterations in IFN1 signaling (Jak-STAT)? During viral infection, there is modulation of these pathways to produce vicious cycle in response to infection.

• Figure 3; “Others….” Please name some in foot note of the figure legend

• Please rewrite all the statements in past tense in results and methodology

• Few errors can be observed in phrase and sentences; Please revise

• Figure 4(g) should be revise; GO terms are too close to read

• In methodology; please detail “Acetylation distribution”

• In methodology, “we defined an enrichment fold change score by (target gene in term * total genes)/(total target genes * gene in term)”; How do you define this? Can you provide a reference for it?

Thank you

**Have the authors made all data and (if applicable) computational code underlying the findings in their manuscript fully available?**

Reviewer #1: Yes

Reviewer #2: Yes

Reviewer #3: Yes

PLOS authors have the option to publish the peer review history of their article (what does this mean?). If published, this will include your full peer review and any attached files.

Reviewer #1: **Yes: **

Reviewer #2: **Yes:**

Reviewer #3: No
---

## [Decision Letter · Decision Letter 1]

13 Jul 2023

Dear Dr. Drier,

Thank you very much for submitting your manuscript "SARS-CoV-2 infection perturbs enhancer mediated transcriptional regulation of key pathways" for consideration at PLOS Computational Biology. As with all papers reviewed by the journal, your manuscript was reviewed by members of the editorial board and by several independent reviewers. The reviewers appreciated the attention to an important topic. Based on the reviews, we are likely to accept this manuscript for publication, providing that you modify the manuscript according to the review recommendations.

Sincerely,

Shailendra S. Gurav, PhD

Guest Editor

PLOS Computational Biology

Rob De Boer

Section Editor

PLOS Computational Biology

Reviewer's Responses to Questions

**Comments to the Authors:**

Reviewer #1: The authors seems to complied the review comments. The revised version has the potential to be published.

Reviewer #2: Nice work of revising the draft.

Reviewer #3: Dear authors, I am still puzzled with enrichment fold change score calculation? You responded in line 426-427 but what is the basis for that? Please quote this calculation if not please detail the rationale for this formula for enrichment fold change score. Further, I do not agree on your statement “We believe that it is important to give credit and provide references when it is due” for my query “Please avoid providing any references in the results section”. The result section in the manuscript (research) is only meant to provide the findings of your methodology hence it is called results section. I did not mean not to give the credit but it should be in the order. It would be better you could discuss the finding and quote the statement only in discussion section not in results.

Thanks

**Have the authors made all data and (if applicable) computational code underlying the findings in their manuscript fully available?**

Reviewer #1: Yes

Reviewer #2: None

Reviewer #3: None

PLOS authors have the option to publish the peer review history of their article (what does this mean?). If published, this will include your full peer review and any attached files.

Reviewer #1: **Yes: **Akash Saggam

Reviewer #2: No

Reviewer #3: No

Figure Files:

Data Requirements:

Reproducibility:

References:

---

## [Decision Letter · Decision Letter 2]

28 Jul 2023

Dear Dr. Drier,

We are pleased to inform you that your manuscript 'SARS-CoV-2 infection perturbs enhancer mediated transcriptional regulation of key pathways' has been provisionally accepted for publication in PLOS Computational Biology.

Best regards,

Shailendra Shivaji Gurav, PhD

Guest Editor

PLOS Computational Biology

Rob De Boer

Section Editor

PLOS Computational Biology

Reviewer's Responses to Questions

**Comments to the Authors:**

Reviewer #3: Thanks for providing reference

**Have the authors made all data and (if applicable) computational code underlying the findings in their manuscript fully available?**

Reviewer #3: Yes

PLOS authors have the option to publish the peer review history of their article (what does this mean?). If published, this will include your full peer review and any attached files.

Reviewer #3: No

---

## [Editor Report · Acceptance letter]

7 Aug 2023

PCOMPBIOL-D-23-00175R2 

SARS-CoV-2 infection perturbs enhancer mediated transcriptional regulation of key pathways

Dear Dr Drier,

I am pleased to inform you that your manuscript has been formally accepted for publication in PLOS Computational Biology. Your manuscript is now with our production department and you will be notified of the publication date in due course.

With kind regards,

Marianna Bach
